# Adherence to Mediterranean Diet, Alcohol Consumption and Emotional Eating in Spanish University Students

**DOI:** 10.3390/nu13093174

**Published:** 2021-09-11

**Authors:** Miguel López-Moreno, Marta Garcés-Rimón, Marta Miguel, María Teresa Iglesias López

**Affiliations:** 1Instituto de Investigación en Ciencias de Alimentación, Consejo Superior de Investigaciones Científicas, Universidad Autónoma de Madrid, 28049 Madrid, Spain; miglop12@ucm.es (M.L.-M.); marta.garces@ufv.es (M.G.-R.); marta.miguel@csic.es (M.M.); 2Grupo de Investigación en Biotecnología Alimentaria, Universidad Francisco de Vitoria, 28223 Madrid, Spain

**Keywords:** university students, alcohol, AUDIT, Mediterranean diet, emotional eater

## Abstract

(1) Introduction: The university period may increase the risk of different unhealthy habits, such as low adherence to the Mediterranean diet, high alcohol consumption and eating in response to specific emotions. The aim of this study was to detect early-risk alcohol consumption and alcohol dependence (ADS), the degree of adherence to the Mediterranean diet and emotional eating in university students of the Madrid community. (2) Methods: For each individual, anthropometric parameters, the Alcohol Use Disorders Identification Test (AUDIT), AUDIT-Consumption (AUDIT-C), the Mediterranean Diet Adherence Screener (MEDAS) and the Emotional Eater Questionnaire (EEQ) were assessed. (3) Results: A total of 584 university students aged 20.5 (sex ratio = 0.39) were enrolled in a cross-sectional study. In total, 63.6% of students showed low adherence to the Mediterranean diet, with no differences by sex (64.3% female versus 61.5% male, *p* = 0.19). According to the AUDIT questionnaire, 26.2% of participants were categorized as high-risk drinkers and 7.7% as ADS. About 38.6% of the students were categorized as eating very emotionally or eating emotionally, and 37.2% were categorized as low emotional eaters. A weak positive correlation was observed between the EEQ and BMI in female students (rho= 0.15, *p* = 0.03). (4) Conclusions: University students in our sample showed a low adherence to the Mediterranean diet, an important high-risk alcohol consumption and low emotional eating.

## 1. Introduction

Excess body weight is a public health problem. It is reported that 39% and 13% of the population are affected by overweight or obesity, respectively [1]. In the university population, overweight prevalence ranges from 16% to 32%, and obesity from 4% to 20% [2,3,4]. Unhealthy habits are a common practice among university students during the university period, such as decreased/or low physical activity, high junk food consumption, evening snacking, high levels of perceived stress, increased workload, dietary restraint, living in residence halls and alcohol consumption [5,6,7]. The transition to university is a critical period for young adults. Facing their first opportunity to make their own food choices, poor culinary skills and the obesity environments with a wide availability of fast food may determine the adoption of unhealthy eating habits and sedentary patterns, leading to increased weight, which may persist during adulthood [8]. For this reason, they are an important target population for the promotion of healthy lifestyles and the reduction in the risk of developing chronic non-communicable disease in adulthood.

The diet of young people is progressively becoming further from the Mediterranean pattern as a consequence of developing unhealthy eating patterns [9]. It is well known that university students show worse Mediterranean diet adherence, with an excess of alcohol, fat and sugar and on the contrary, a low intake of fruit, vegetables and pulses [10]. The Mediterranean diet represents a dietary pattern that incorporates healthy traditional eating habits of populations from countries surrounding the Mediterranean Sea. The Mediterranean diet promotes primarily unprocessed foods, including a high consumption of vegetables, fruit, whole grains, legumes, nuts, beans, fish and olive oil, and the low consumption of red meat and dairy products [11]. The traditional Mediterranean diet has been associated with a reduced risk of several chronic diseases, such as type 2 diabetes mellitus and cardiovascular disease, and with a reduction in cognitive damage [12]. Moreover, the Mediterranean diet has beneficial effects on life expectance and quality of life [13]. On the other hand, alcohol often plays an essential role in young people’s lives when they start university, a consumption pattern that is characterized by large amounts being drunk in short periods of time following periods of abstinence, also known as binge drinking (intake of ≥4 alcoholic beverages in a short period of time) [14]. This excessive alcohol consumption deviates from the Mediterranean dietary pattern and has important health implications, increasing the risk of brain damage and neuropsychological repercussions; its long-term consequences on the risk of cancer are broadly described [15,16]. Early alcohol consumption is related with abuse and alcohol dependence problems [17]. Therefore, public health measures, especially for young adults, are necessary to reduce alcohol use.

In general terms, it is known that emotions are really important in food election and negative moods such as stress, anxiety or psychiatric disorders are linked with obesity and alcohol abuse risk [18,19]. Emotional eating is eating in response to negative emotions, which increases the risk of excessive energy intake [20]. A meta-analysis review shows that obesity increases the risk of depression and depression also acts as a risk factor for obesity [21]. People with depressive symptoms show a tendency to consume “comfort food”, which means high-energy-dense foods, such as sweet high-fat foods [22]. In a previous study, a higher-energy-dense diet was related to a 2.56-fold increased risk of stress [23]. In addition, it was suggested that stressful situations such as home confinement due to COVID-19 can increase the risk of emotional eating [24]. Similarly, it has been observed that eating behavior shows gender differences, as women seem to have a greater restrained and emotional eating tendency [25,26].

The research question for this study was: is adherence to the Mediterranean diet, alcohol consumption and/or emotional eating associated with the weight status in university students? 

The aim of this study was to assess the prevalence of alcohol abuse, emotional eating and adherence to the Mediterranean diet, in addition to its possible implications for body weight status in Spanish university students. 

## 2. Material and Methods

A cross-sectional observational study was carried out during the academic year 2018–2019 from February to May 2019 in Madrid, Spain. Prior to the start of the study, each participant was informed about the purpose of the study and written informed consent was obtained from all subjects. Students’ participation was voluntary, and the survey was anonymized. The sample included students from different degrees in health sciences: Medicine, Pharmacy, Biotechnology, Nursing, Physiotherapy, Psychology and Gastronomy. Data collection was carried out through online forms and only participants with complete information were included in the present study. This study was conducted according to the guidelines laid down in the Declaration of Helsinki and all procedures were reviewed and approved by the ethics committee at the Universidad de Francisco de Vitoria (36/2017).

The anthropometric assessment was conducted using calibrated digital scales SECA^®^ 840 and 877 (SECA Vogel and Halke, Hamburg, Germany) and portable stadiometers SECA^®^ 214 and 217 (SECA Vogel and Halke, Hamburg, Germany). The measurement of the participants’ body weight was carried out in light clothes and barefoot in kilograms, to the nearest 100 g unit (0.1 kg), and height was assessed with participants standing fully erect and feet together, head in the Frankfort plane and arms hanging freely to the nearest millimeter (0.1 cm). Body mass index (BMI) was calculated according to the formula weight (kg)/height (m)^2^. Participants were categorized as underweight (BMI < 18.5 kg/m^2^), normal weight (BMI 18.5–24.9 kg/m^2^), overweight (BMI 25–29.9 kg/m^2^) and obese (BMI ≥ 30 kg/m^2^) [27].

Validated questionnaires were administered and answered by each participant individually: the Mediterranean Diet Adherence Screener (MEDAS), the Alcohol Use Disorders Identification Test (AUDIT) and the Emotional Eater Questionnaire (EEQ).

In this study, to determine adherence to the Mediterranean diet, we used PREDIMED scores obtained by the MEDAS which was originally developed for the Spanish population [28]. This questionnaire contains 14 items representing the dietary components of the Mediterranean diet. Each item was scored as 0 (no adherence to a dietary component) or 1 (maximal adherence to a dietary component). The range of index was 0–14. We used cut-offs to classify subjects as high (>9), moderate (7–9), or low adherers (0–7) in each of the components of an MD.

The Alcohol Use Disorder Identification Test (AUDIT) was included to assess the drinking habits of the university students [29]. The questionnaire was designed by the World Health Organization to identify individuals with risky drinking habits through a simple screening test. The questionnaire contains 10 items, 8 of which are on a Likert scale of 5 categories ordered from 0 (never/1 or 2 units) to 4 (daily/10 or more units). The two remaining items also use a Likert scale but with 3 categories ordered from 0 to 2. The validation of the test on a Spanish university population by García Carretero et al. found the internal consistency of the AUDIT test to be α = 0.75 [30]; they proposed adequate cut-off points for participants who were low-risk drinkers (score of 0–7 for men and 0–5 for women), participants who were high-risk drinkers (score 8–12 for men and 6–12 for women) and participants who had probable alcohol dependence syndrome (ADS) (score ≥13 for both sexes). A short version, AUDIT–Consumption (AUDIT-C) consists of the first 3 AUDIT items, including the frequency and amount of alcohol consumption and the frequency of binge drinking. AUDIT-C cut-off of ≥3 in women and ≥4 in men was used to identify hazardous drinking [31]. Abstainers were taken into account in the study; these students responded that they never drink alcohol.

A ten-item questionnaire called the Emotional Eater Questionnaire (EEQ) was created to assess to what extent emotions affect eating behaviors. The EEQ provides information on moods in emotional eaters with respect to emotional regulation for eating and the subsequent implementation of more effective programs to lose weight. This questionnaire is composed of 10 items with four categories: (1) never, (2) sometimes; (3) generally and (4) always. The global score ranges from 0 to 30. Participants were classified as: non-emotional eaters (scores of 0–5), low emotional eaters (scores of 6–10), emotional eaters (scores of 11–20) and very emotional eaters (scores of 21–30) [32].

Descriptive outcomes are presented as the mean (standard deviation) and the categorical variables are presented as frequencies of occurrence (n) and percentages (%). The Shapiro–Wilk test was performed to evaluate the distribution of variables. The two-sample Student’s t test (with Levene’s test for equality of variances) was used to compare two means. Differences between groups were tested using the Chi Squared test and one-way ANOVA. Non-parametric methods were used, such as the Mann–Whitney test in the case of two categorical groups. Cohen’s d and partial eta squared were calculated to estimate the effect size for t-tests and for ANOVA, respectively. In the non-parametric tests, the effect size was performed with the probability of superiority (PS_est_) [33].The Spearman correlation coefficient was used to assess the correlation between variables. The correlation coefficients were interpreted using the following thresholds: trivial (< 0.1), small (0.1–0.39), moderate (0.4–0.69), strong (0.7–0.89) and extremely perfect (≥0.9) [34]. The level of significance was set at *p* < 0.05. Statistical analysis was performed with SPSS^®^ version 22.0 (SPSS, Chicago, IL, USA).

## 3. Results

The 682 students enrolled in different degrees in health sciences were invited to participate voluntarily in the study and were informed of the procedure. Of these 682 students, 584 agreed to participate and 98 students were excluded due to missing data. Finally, the studied sample was 584 students: 164 males (28.1%) and 420 females (71.9%); sex ratio = 0.37. Table 1 shows student characteristics and anthropometry; 26.7% showed overweight or obesity (29.4% male 27.1% female, *p* = 0.62). 

Table 2 summarizes different items of MEDAS by sex. The average score obtained from the MEDAS questionnaire was 7.71 ± 1.92, being higher in men (7.98 ± 1.93) compared to women (7.71 ± 1.92) (Mann–Whitney test, PS_est_ = 0.44, *p* = 0.03). In this respect, 63.6% of students showed low adherence to the Mediterranean diet, with no differences by sex (64.3% female versus 61.5% male, *p* = 0.19). Adherence to the Mediterranean diet was similar in participants with normal weight and participants with overweight or obesity (Chi Square test, *p* > 0.05). Comparing items of MEDAS reported that a high percentage of the students did not reach the intake recommendation of nuts, fruits, legumes and fish consumption. In addition, women with a normal weight were more compliant with the fish consumption recommendation than women with overweight or obesity (43% vs. 33%, respectively, Chi Square test, V = 0.09, *p* = 0.04).

Regarding the pattern of alcohol consumption, the median AUDIT score was 5.3 ± 4.4, with a higher score for women than for men (5.3 ± 4.6 vs. 5.0 ± 4.0, respectively) (Mann–Whitney test, *p* > 0.05). Table 3 shows results from the AUDIT and AUDIT-C questionnaires. In total, 27% reported drinking alcohol 2–3 times/week, 42% reported drinking alcohol 2–4 times/month, and 8.6% reported being abstainers (with no difference between sex, *p* > 0.05, for all). According to the AUDIT-C questionnaire, a higher proportion of women showed high-risk consumption compared to men (62.6% and 74.2%, respectively) (Chi Square test, χ^2^ = 7.05, V = 0.11 *p* = 0.008) (Table 3). Men with high-risk alcohol consumption showed lower adherence to the Mediterranean diet (Chi Square test, χ^2^ = 4.78, V = 0.17, *p* = 0.02). There was no difference in alcohol consumption between students with normal weights and those with overweight or obesity (*p* > 0.05).

The average score for the emotional eating questionnaire was 10.1 ± 6.0 (men 10.0 ± 6.1 and women 10.1 ± 6.9). Table 4 shows proportions of students scoring as low emotional eaters, emotional eaters and very emotional eaters. In total, 38.6% of students were classified as emotional or very emotional eaters according to EEQ (Table 1).

An association was found between BMI and EEQ. Specifically, students with obesity had a higher risk of being very emotional eaters (9.8%) compared to students with overweight (4.4%) or normal weight (7.2%) (Chi Square test, V = 0.11, *p* = 0.02). Among students affected by obesity, women were found to be more emotional eaters or more very emotional eaters than men (25% vs. 48.3%, respectively) (Chi Square test, V = 0.46, *p* = 0.03). In addition, a weak positive correlation was observed between the EEQ and BMI in female students (rho= 0.15, *p* = 0.03). In relation to the AUDIT, it was found that those participants with ADS reported eating more when they were stressed, angry or bored than those without risky consumption (33.3% vs. 10.7%, respectively) (Chi Square test, V = 0.14, *p* = 0.04).

## 4. Discussion

In this study, we evaluated the prevalence of the adherence to the Mediterranean diet, the drinking habits and the emotional eating prevalence in a sample of a university students in Madrid. Based on the weight status, we reported that 27% of the study population presented a BMI classified as overweight/obesity without differences by sex. These data are in accordance with the values reported in others Spanish university populations [35,36]. In a study in Spanish University students, the overweight/obesity prevalence was 25.8% and no differences were observed between women and men [37]. Furthermore, in our study, the prevalence of university students with underweight (12%) was similar to previous studies [38,39,40]. Teleman et al. found that the prevalence of underweight was 14% for Italian university students, being higher among women (19%) than men (2%) [41].

A non-Mediterranean pattern, with the intake of unhealthy foods, increases the risk of inflammatory and oxidative stress linked with cardiovascular diseases, obesity and cancer [42]. A systematic review of 41 observational studies revealed that high adherence to the Mediterranean diet is associated with an improvement in all parameters of metabolic syndrome in adults [43]. The present study shows that a large proportion of university students had low adherence to the Mediterranean diet. Cobo-Cuenca et al. reported that 66% of students had low adherence to the Mediterranean diet, similar with 64% observed in our sample [37]. These findings are in line with those reported by other previous studies in European university students [44]. Moreover, we observed lower adherence with no significant differences in females (36%) than in males (39%), a finding that follows the trend shown in previous studies [37]. 

The impact of Mediterranean diet adherence on weight status has shown the heterogeneity of results. Low adherence to the Mediterranean diet has been associated with higher BMI, body fat and visceral fat percentages [45]. In contrast, other studies have not found a relationship between adherence to the Mediterranean diet and different anthropometric markers [46,47,48]. In our sample, the relationship of adherence to the Mediterranean diet with the risk of overweight/obesity could not be observed either.

On the other hand, we observed that a low proportion of students (39% both in sexes) ate three servings of fish or shellfish/week. In previous studies, lower fish consumption was seen in the university population [49,50]. As a matter of fact, Cobo-Cuenca et al. reported that only 30% of male and 32% of female Spanish university students had fish and shellfish in agreement with the Mediterranean diet recommendation [37]. Female university students with overweight or obesity showed lower adherence to weekly fish consumption recommendations. A large dose–response meta-analysis of prospective studies found that fish consumption is associated with a reduced risk of abdominal obesity [51]. This beneficial effect on body composition could be attributed to their omega-3 polyunsaturated fatty acid content [52], protein [53] and vitamin D [54].

The median AUDIT score of 5.3 ± 4.4 observed in the present study is similar to a previous report in Spanish university students [55], but lower than that observed in other European countries (with the median AUDIT of score 6.9 ± 5.5) [56]. According to the risk categories, more than 30% of the university students had high-risk consumption or ADS. Heather et al. reported a 10% prevalence of ADS in English university students; however, in this study, the cut-off points used to identify ADS (≥20) was different from the present study (≥13) [57]. Risky drinking has been linked to impaired mental health among university students, reduced satisfaction with life as well as increased emotional and social loneliness [58,59]. Cognitive performance is crucial during the university stage; numerous previous studies have shown that alcohol consumption has a negative impact on academic performance [36,60]. This may be because excessive alcohol consumption has also been linked to impaired metabolic flexibility and memory impairment [61,62]. Moreover, our results are relevant when considering the relationship between a high AUDIT score and increased mortality [63].

In this study population, it was observed that among male university students with a high-risk alcohol consumption, there was low adherence to the Mediterranean diet. A pattern of excessive alcohol consumption has been linked to a poor quality diet with the lower consumption of many different food groups included in the Mediterranean diet [64]. However, in contrast to previous studies, there was no evidence of a relationship between BMI and alcohol abuse [65,66]. This diversity of the results may be partly due to the different methods of alcohol assessment used, as well as the study population analyzed.

In the present study, the median EEQ was 10.1 ± 6.0, which is lower than previous findings. López-Guimerà et al. observed mean values of EEQ 11.8 ± 6.0, but in this study, the population was 39 ± 12 years. In addition, 38.6% of students were classified as emotional eaters or very emotional eaters [67]. These results are consistent with a previous study carried out during the period of home confinement due to COVID-19 in the Spanish population [24]. For many people, eating in response to negative emotions or stress is quite common. Individuals with depressive moods are susceptible to consume “comfort food” to help control their negative moods [68].

Eating in response to mood can be problematic, as shown by studies that have linked emotional eating to BMI [69,70]. In this work, there was a weak positive correlation between emotional eating and BMI (r = 0.15; *p* ≤ 0.05) in female students. Işik and Cengiz also found a positive correlation between emotional eating and BMI without gender differences [71]. Other studies used different questionnaires to measure emotional eating, such as the Eating and Appraisal Due to Emotions and Stress Questionnaire (EDADES) and the Three-Factor Eating-R18 (TFEQ-R18), relating strong association between weight gain and negative emotions [18,72]. Body dissatisfaction, particularly among women, has been linked to different eating disorders, such as restrained eating and orthorexia nervosa [73]. This suggests the coexistence of weight disorders and unhealthy relationships with food, which negatively affects mental health throughout one’s lifespan [74].

As limitations, firstly, the study was conducted in young adults within the geographical area of the Spain (urban part of the northwest area of Madrid). Secondly, we did not have data for real alcohol consumption; nevertheless, the AUDIT test seems to be a good instrument to detect alcohol risk behavior. Thirdly, questionnaires were self-administered. Fourthly, there was an imbalance between sexes, which could reduce the representativeness of the sample. However, it is important to note that the majority of Spanish university students in health sciences are women [4,75]. Finally, the MEDAS questionnaire may be more appropriately used to measure adherence to the Mediterranean diet in adults, because young students normally do not drink wine.

The main strength of this study lies in the relationship between different variables (adherence to the Mediterranean diet, alcohol consumption and emotional eating), as few studies to date have associated these variables in the university population.

## 5. Conclusions

In conclusion, this study showed a large proportion of students with low adherence to the Mediterranean diet, and a high proportion of students with high-risk alcohol consumption and eating in response to emotions. Future co-design work to design invention strategies focusing on eating behaviors and alcohol use in young adults must be developed. Furthermore, to avoid serious unhealthy consequences, the implementation of actions to correct deviations of alcohol abuse and negative food relationships are necessary.

## Figures and Tables

**Table 1 nutrients-13-03174-t001:** Descriptive characteristics of the study sample by sex.

	All (*n* = 584)	Men (*n* = 164)	Women (*n* = 420)	*p*
Age (years)	21.2 (SD 4)	22.7 (SD 5)	21.0 (SD 4)	0.03
Height (m)	1.68 (SD 0.09)	1.71 (SD 0.09)	1.67 (SD 0.09)	0.00
Weight (kg)	63 (SD 12)	64 (SD 12)	62 (SD 12)	0.14
BMI (kg/m^2^)	22.3 (SD 4.5)	22.0 (SD 4.6)	22.4 (SD 4.5)	0.38
BMI				0.62
Underweight (%)	11.8	13.3	11.2	
Normal weight (%)	60.5	57.3	61.7	
Overweight (%)	20.1	21.0	19.8	
Obesity (%)	6.6	8.4	7.3	
Adherence Mediterranean diet				0.19
Low adherence (%)	63.6	61.5	64.3	
Good adherence (%)	36.4	38.5	35.7	
Total Mediterranean diet score	7.7 (SD 1.9)	7.8 (SD 1.9)	7.6 (SD 1.9)	0.03

Values are the mean and SD. *p*-value by Chi Square test for categorical variables, Mann–Whitney test and Kruskal–Wallis test for continuous variables among two or more groups, respectively. BMI = body mass index. Adherence Mediterranean diet score: low adherence (score 1–8) or good adherence (score ≥ 9).

**Table 2 nutrients-13-03174-t002:** Agreement with the recommendations based on each item of the Mediterranean Diet Adherence Screener (MEDAS-14 item score), by sex.

	Recommendation	Agreement with the Recommendation		No Agreement with the Recommendation	
Questions from the MEDAS-14		**Men**	**Women**	**Men**	**Women**
1. Do you use olive as a main culinary fat?	Yes	98	95	2	5
2. How much olive oil do you consume in a given day (including oil used for frying, salads, out-of-house meals, etc.)?	≥4 tbsp	67	62	33	38
3. How many vegetables servings do you consume per day? (1 serving; 200 g (consider side dishes as half a serving))	≥2 (≥1 portions raw or as a salad)	56	57	44	43
4. How many fruits (including natural fruit juices) do you consume per day?	≥3	41	38	59	62
5. How many servings of red meat, hamburger, or meat products (ham, sausage, etc.) do you consume per day? (1 serving: 100–150 g)	<1	65	60	35	40
6. How many servings of butter, margarine, or cream do you consume per day? (1 serving: 12 g)	<1	71	72	29	28
7. How many sweet or carbonated beverages do you drink per day?	<1	**68**	**61**	**32**	**39**
8. How much wine do you drink per week?	≥7 glasses	**9**	**4**	**91**	**96**
9. How many serving of legumes do you consume per week? (1 serving: 150 g)	≥3	34	39	66	61
10. How many servings of fish or shellfish do you consume per week? (1 serving 100–150 g of fish or 4–5 units or 200 g of shellfish)	≥3	39	39	61	61
11. How many times per week do you consume commercial sweets or pastries (not homemade), such as cakes, cookies, biscuits, or custard?	<3	54	54	46	46
12. How many servings of nuts (including peanuts) do you consume per week? (1 serving 30 g)	≥3	29	29	71	71
13. Do you preferentially consume chicken, turkey, or rabbit meat instead of veal, pork, hamburger or sausage?	Yes	71	72	29	28
14. How many times per week do you consume vegetables, pasta, rice, or other dishes seasoned with sofrito (sauce made with tomato and onion, leek, or garlic and simmered with olive oil?	≥2	76	78	24	22

Values presented as percentages. *p* by Chi Square test. Bold values indicate statistical significance different between genders (*p* < 0.05.). Tbsp = tablespoon. Recommendation according to Martínez-González et al. [22].

**Table 3 nutrients-13-03174-t003:** Problems with alcohol consumption by sex according to AUDIT.

	Men *N* (%)	Women *N* (%)	All *N* (%)	*p*-Value
AUDIT				0.002
Low-risk drinker	115 (70.1)	271 (64.5)	386 (66.1)	
High-risk drinker	29 (17.7)	124 (29.5)	153 (26.2)	
Drinker with probable ADS	20 (12.2)	25 (6.0)	45 (7.7)	
AUDIT-C				0.008
Low-risk drinking	121 (74.2)	263 (62.6)	384 (65.9)	
Hazardous drinking	43 (26.0)	157 (37.4)	200 (34.1)	

Chi Square test, *p* < 0.05. AUDIT (Alcohol Use Disorder Identification Test (AUDIT); AUDIT-C (Alcohol Use Disorder Identification Test–Consumption (AUDIT-C); ADS (Alcohol Dependence Syndrome). AUDIT cut-off: low-risk drinker (score 0–7 for men and 0–5 for women), high-risk drinker (score 8–12 for men and 6–12 for women) and probable alcohol dependence syndrome (ADS) (score ≥ 13). AUDIT-C cut-off: low-risk drinking (score <3 in women and <4 in men) and hazardous drinking (score ≥3 in women and ≥4 in men).

**Table 4 nutrients-13-03174-t004:** Emotional eating by sex according to EEQ.

	All (*n* = 584)	Men (*n* = 164)	Women (*n* = 420)	*p*
No emotional (%)	24.3	21.0	24.2	0.81
Low emotional eater (%)	37.2	40.6	35.4	
Emotional eater (%)	32.4	31.5	33.9	
Very emotional eater (%)	6.2	7.0	6.5	

*p*-value by Chi Square test for categorical variables; emotional eating score: non-emotional eater (score 0–5), low emotional eater (score 6–10), emotional eater (score 11–20) and very emotional eater (score 21–30).

## Data Availability

Publicly available datasets were analyzed in this study. This data can be found here: 10.6084/m9.figshare.16601543.

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
