# Peer review of "Adherence to Mediterranean Diet, Alcohol Consumption and Emotional Eating in Spanish University Students"

_nutrients, 2021, doi:10.3390/nu13093174_

Round 1
Reviewer 1 Report
please see attachment for details.
Review
Summary
In the current manuscript entitled Alcohol consumption, adherence to Mediterranean diet and emotional eater study in university students, the investigators present data from university students from different degrees in health sciences in Madrid, Spain. The study is a cross-sectional observational study carried out during the academic year of 2018-2019. In this report, the authors seek to analyze harmful alcohol consumption, the degree of adherence to the Mediterranean diet, and emotional eating among Spanish university students. In doing so the investigators present that one third of the university students had harmful alcohol consumption pattern, two thirds had low adherence to the Mediterranean diet, and nearly 40% were eating in response to emotions. Their results give additional
insight in alcohol- and eating habits, and problems related to eating behavior in the Spanish undergraduate population. Underscoring both the need to better understand the reasons for unhealthy habits, and the need for implementation of educational programs to promote healthy habits among university students at risk.
The manuscript would benefit from consistency in the order of the three areas of health research alcohol consumption, adherence to Mediterranean diet and emotional eating. Deciding the order and following the same pattern consistently throughout would make it
easier for the reader to follow.
What follows is a list of the inconsistency of the order of the three assessed areas of health research:
Title:
• 1) Alcohol consumption, 2) Adherence to the Mediterranean diet, 3) Emotional eating.
Abstract:
• Introduction: 1) Adherence to the Mediterranean diet, 2) Alcohol consumption, 3) Emotional eating.
• Aims: 1) Alcohol consumption, 2) Adherence to the Mediterranean diet, 3) Emotional eating.
• Methods (anthropometrics), 1) Alcohol (AUDIT), 2) Adherence to the Mediterranean diet (MEDAS), 3) Emotional eating (EEQ).
• Results: 1) Adherence to the Mediterranean diet, 2) Alcohol consumption, 3) Emotional eating.
• Conclusion: 1) Adherence to the Mediterranean diet, 2) Alcohol consumption, 3) Emotional eating.
Introduction:
• (Body weight/anthropometry), 1) Adherence to the Mediterranean diet, 2) Alcohol consumption, 3) Emotional eating.
Materials and Methods:
• (Anthropometrics), 1) Adherence to the Mediterranean diet (MEDAS), 2) Alcohol use disorder identification test (AUDIT), 3) Emotional Eating Questionnaire (EEQ), (statistics).
• Aims of the study at the end of introduction, 1) Alcohol consumption, 2) Emotional eating, 3) Adherence to the Mediterranean diet.
Results:
• The text covers as follows: (enrollment numbers and anthropometrics), 1) Adherence to the Mediterranean diet (MEDAS), 2) Alcohol consumption (AUDIT), 3) Emotional eating (EEQ).
Figure 1.
• Flow chart: 1) AUDIT, 2) MEDAS, 3) EEQ
Table 1.
• (Demography/anthropometrics), 1) Adherence to the Mediterranean diet (MEDAS), 2) Emotional eating (EEQ).
Table 2.
• Adherence to the Mediterranean diet (MEDAS)
Table 3.
• 1) Alcohol consumption (AUDIT & AUDIT-C)
The title:
“Alcohol consumption, adherence to Mediterranean diet and emotional eater study in university students”, could be a little bit more descriptive and precise, consider suggestion: “Alcohol consumption, adherence to the Mediterranean diet and emotional eating in Spanish university students”
Abstract:
The abstract is clear and relevant. As mentioned earlier, deciding the order of the three areas of health research in this manuscript and following the same pattern consistently should also be reflected in this part. The abstract would further benefit from a clear conclusion.
Overall:
• Results repeated in the abstract would look cleaner, and be easier to read, without decimals.
Line 10 and 18: “alcohol dependence (SDA)” should be “alcohol dependence syndrome (ADS)”. Line 18: Use only abbreviation ADS, without parentheses, as it has been explained once before in the abstract.
Line 13: Include “and AUDIT-consumption (AUDIT-C)” after “(AUDIT)” as both questionnaires were used to detect early risk alcohol consumption.
Line 14: Insert “(EEQ)” after “Emotional Eater Questionnaire”.
Line 18-19: Transitional words and phrases: Starting the sentence with “Finally, “ usually have a “Firstly, “ and “Secondly, “ etc. describing the findings before. I suggest skip it.
Consider following for Line 18-19: “About 40% of the students were categorized as eating very emotionally or eating emotionally, and 35% were categorized as low emotional eaters.”
Line 22-24: The manuscript would further benefit from a clear conclusion. What do the authors mean with this conclusion? “... to explore the beliefs and behaviors...” ?
Introduction:
The introduction is well written, condensed and with a good flow. As mentioned earlier, deciding the order of the three areas of health research in this manuscript and following the same pattern consistently should also be reflected in this part. Suggestions are on including
additional information for clarification.
Line 35-37: Although the authors have cited previously published reporting about unhealthy habits (line 30) and unhealthy eating pattern (line 35) in university student, a brief description regarding the unhealthy habits/eating pattern of university students should be included in the current report (i.e. decreased/or low physical activity, high junk food consumption, evening snacking, high levels of perceived stress, increased workload, dietary
restraint and living in residence, and alcohol consumption).
The introduction would further benefit from a description of the student years/adolescent period of life (i.e. Transition to university is a critical period for young adults. Facing their first opportunity to make their own food choices, having lack of time and money may determine the adoption of unhealthy eating habits and sedentary patterns, leading to increased weight,
which may persist during adulthood).
Also, a brief overview of what the Mediterranean diet includes or excludes, to help get the reader on track. (i.e. The Mediterranean diet represents a dietary pattern that incorporates healthy traditional eating habits of populations from countries surrounding the Mediterranean Sea. The Mediterranean diet promotes primarily unprocessed foods, including a high consumption of vegetables, fruit, whole grains, legumes, nuts, beans, fish, olive oil, and low consumption of red meat and dairy products).
Line 58: “energetic foods” suggestion of rephrasing to “high-energy-dense foods”
Materials and Methods:
As mentioned earlier, deciding the order of the three areas of health research in this manuscript and following the same pattern consistently should also be reflected in this part.
Suggestions for clarification as follows:
Line 84: “(WHO)” is not needed, as it is not appearing again.
Line 94: Please insert citation/ref for cut-offs.
Line 96: Please insert citation/ref for AUDIT.
Line 101-104: According to the English translated abstract of reference 23, Garcia Carretero et al. (23), cut-off points for high-risk drinking in students are 8 for male and 6 for female (and 13 for probable ADS, in both sexes). Did the authors use the different cut-offs for the
AUDIT between sexes? If so, please clarify in methods. Also, according to the abstract (23) AUDIT-C cut-off points for high-risk patterns for male and female are 5 and 4, respectively. Where these used?
Line 102 and 103: Citation style differs from the rest, keep one style for consistency.
Line 103-104: Abstainers are not mentioned in Materials and Methods, but are mentioned in Results line 172. This category should also be explained in Materials and Methods and that they are withdrawn from the calculation of the proportion of students with risky alcohol consumption.
Line 107: Revise sentence, suggestion: “Adequate cut-off for AUDIT-C, ≥3 for detecting hazardous drinking, and ≥4 for diagnosable disorders (24).”
Line 109: EEQ is used here for the first time and should be explained before. Emotional Eating Questionnaire (EEQ)
Line 113: “We classified participants” should be “Participants were classified as:”
Line 115: Citation/ref to classification.
Line 116-117: The statistics presented on the validity of EEQ in Spanish university students, from the cited report Bernabéau et al. (26), are results from the validation of the EEQ-7 item version. According to line 111 the 10-item questionnaire was used. Please revise.
Bernabéau et al. (26): “The study presents a new validation of the EEQ, resulting in a seven items one-factor instrument, easy to understand and to administrate to large samples of non-clinical people, valid and reliable in evaluating the degree of emotion in relation with food intake in a university population.”...“Cronbach’s Alpha of the one-dimension test based on 7 items was 0.753, and convergent validity r=0.317;
p<0.001”... “Finally, 7-items EEQ score was not predictive of BMI among university students.”
Line 130: Citation/ref to thresholds.
Results:
As mentioned earlier, deciding the order of the three areas of health research in this manuscript and following the same pattern consistently should also be reflected in this part. In what hat follows, a few suggestions on revision of table/text content, and suggestions on
corrections according to interpretation of cited references, and suggestions to consider rewriting:
Overall:
• References may be in any style, provided that you use the consistent formatting throughout.
• Acronyms and abbreviations should be defined the first time they appear in each of three sections: the abstract; the main text; the first figure or table. When defined for the first time, the acronym/abbreviation should be added in parentheses after the written-out form.
• Chose to call the sexes men and women or male and female, for consistency.
• Chose to write data with the same amount of decimals, for consistency.
Line 141-145: The three first sentences can be erased as they repeat data seen in the table.
Please correct age data to one decimal and with correct SD as written in the first sentence.
Suggestion of what to leave/rewrite: “Table 1 shows patient characteristics and anthropometry, 26.7% were overweight or had obesity (29.4% male and 27.1% female, p>0.05).”
Table 1.
• According to text in (Line 112-113) in section Materials and Methods the authors classified participants accordingly (from a global score range 0 to 30): non-emotional eater (score 0-5), low emotional eater (score 6-10), emotional eater (score 11-20), and very emotional eater (score 21-30). Please use the same classifications in Table 1 for consistency.
• It would also be helpful to have the scores in the table or table text.
• According to the table text bold values indicate statistical significance. There are no bold values.
• P-values are shown for age, height, weight, BMI, BMI for underweight, low adherence to MD, and no emotional eating. The rest of the data have no p-values.
Does this mean that the data not followed by p-value have the same p-value as the data above? Please clarify.
• According to the discussion, line 215-216 “observed less adherence” if this is not significant it should be stated in the discussion.
• The data in Table 1 is not consistent with the results presented in abstract Line 18-19, and with results in text Line 183-185. In the abstract “39.8% of students were very emotional or emotional eater and 35.4% were low emotional eater”, and in the discussion (line 255-256) “38.6% of students were classified as emotional eaters or very emotional eaters”. Please revise all.
Line 154-155: Change sentence to person first language: “Adherence to the Mediterranean diet was similar in participant with normal weight and participants with overweight or obesity (Chi square test, p>0.05).”
Line 158: Change sentence to person first language: “Women with normal weight were more compliant with the fish consumption recommendation than women with overweight or obesity (43% vs. 33%, respectively, Chi-square test, V=0.09, p<0.05).”
Table 2.
• Insert citation/reference to the recommendations, cut-offs for the different recommendations according to MEDAS-14.
• Only question 7 and 8 are bold, is that correct?
Line 167-170: Repeated information as presented in table 3 and can be erased. Instead after the first sentence: “Table 3 shows results from the AUDIT and AUDIT-C questionnaires.
Twenty-seven percent reported drinking alcohol 2-3 times/week, 42% reported drinking alcohol 2-4 times/month, and 8.6% reported being abstainers (with no difference between sex, p>0.05, for all).”
Line 174-176: Repeated information as presented in table 3 and can be erased, or only keep text (without data) and refer to Table 3. If kept: Line 174-175: “higher” should be “lower” and “high-risk consumption” should be “low-risk consumption”, according to results in Table 3.
Line 176: Transition word “Similarly” I suggest, skip it. Consider rewriting sentence accordingly: “Men with high-risk alcohol consumption had lower adherence to the Mediterranean diet (Chi square test, X2=7.05, V=0.11, p<0.05) (Table 3).”
Line 178: Change sentence to person first language: “There was no difference in alcohol consumption between students with normal weight and those with overweight or obesity (p>0.05).”
Table 3.
• Explain/insert cut-offs for AUDIT and AUDIT-C, preferably in the table or table text.
According to Materials and Methods (line 107) and cited Reinert et al. (24), AUDIT-C cut-off ≥3 is for detecting hazardous drinking, and ≥4 is for diagnosable disorders. Are these interpreted as “≥3 = low-risk consumption” and “≥4 = high-risk consumption”?
If so, this should be explained in Materials and Methods. If not, revise text and table to correct terms/interpretations.
• P-values are shown for AUDIT Low-risk drinkers and AUDIT-C Low-risk consumption.
The rest of the data have no p-values. Does this mean that the data not followed by p-value have the same p-value as the data above? Please clarify.
• Abbreviations: Alcohol Use Disorder Identification Test (AUDIT), and Alcohol Use Disorder Identification Test - Consumption AUDIT-C should be explained under the
table.
Line 182-185: Keep first sentence. Second sentence is a repetition of what is presented in table 1. Suggestion after first sentence: “Table 1 shows proportions of students scoring low emotional eaters, emotional eaters, and very emotional eaters. 3X.X% of students were
classified as emotional or very emotional eaters according to EEQ (Table 1.).”
Line 186: Skip the first transitional word “Finally” and start the sentence with “An association...”.
Line 191-192: A correlation value of 0.15 shows there is a positive correlation between two variables, but it is weak and likely unimportant, this should be outlined. Line 19-20, 191-192, 262-263
Discussion:
The discussion is overall relevant with relevant citing and references within the Mediterranean university student population. However, some suggestions on grammatical changes, topics of discussion, interpretation of cited references, and suggestions on to
consider rewriting. Suggestions as follows:
Overall:
• References may be in any style, provided that you use the consistent formatting throughout.
• Acronyms and abbreviations should be defined the first time they appear in each of three sections: the abstract; the main text; the first figure or table. When defined for the first time, the acronym/abbreviation should be added in parentheses after the written-out form.
• Results repeated in the discussion would look cleaner, and be easier to read, without decimals.
Line 200: “date” should be “data”
Line 201: “others” should be “other Spanish university student populations”.
Line 203: Transitional phrase “In the same way”should be “Furthermore”
Line 204: Authors have missed to write “students” after “university” before “with”.
Line 210: “an improved” should be “improvement” plus insert “in adults” at the end of the sentence.
Line 211: Sentence correction suggestion: “The present study shows that a large proportion of university students had low adherence to the Mediterranean diet.”
Line 212-214: Citation style differs from the rest, keep one style for consistency, consider: “Cobo-Cuenca et al. reported the... ... the Mediterranean diet (30), similar with....”
Line 219: “haven ́t” should be “have not”.
Line 223: “(39% men and 39% women” should be “(39% both in sexes)”
Line 225-227: Sentence correction, and citation style differs from the rest, keep one style for consistency: “As a matter of fact, Cobo-Cuenca et al. reported that only 30% male and 32% female Spanish university students had fish and shellfish in agreement with the
Mediterranean diet recommendation (30).”
Line 228: Change sentence to person first language: “female university students with overweight or obesity...”
Line 233: “as” should be “than”
Line 234: Citation style differs from the rest, keep one style for consistency, consider: “... by Cooke et al. (48), and...” Or consider following for Line 233-234: “The median AUDIT score of 5.3±4.4 observed in the present study, is similar to a previous report in Spanish
university students (49), but lower than observed in other European countries
(with median AUDIT score of 6.9±5.5) (48).”
Line 235: Please insert citation/reference 23 in“According to risk categories (23), more than...”
Line 236-237: This statement is incorrectly cited/referenced. In the study by Heather et al.,
(2011) 40% of adolescents were hazardous drinkers (AUDIT score 8-15), 11% harmful drinkers (AUDIT score 16-19), and 10% probable dependence (ADS) (AUDIT 20 or over).
In the present study 26.2% scored high-risk drinkers (AUDIT score 8-12) and 7.7% scored probable dependence (ADS) (AUDIT score 13 and over). Over 30% (33.9%) of the university students in this present study had high-risk consumption and/or ADS (Audit score 8 and above). If correctly compared with the study by Heather et al. AUDIT scores of 8 and above would be 61% of the adolescents. If the authors meant to compare the prevalence of adolescents with probable dependence (ADS), 7.7% in the present study with 10% by Heather et al. The cut-off points for probable dependence (ADS) assessed by AUDIT scores of
13 and above vs. 20 and above would have to be addressed and discussed.
On that note, AUDIT is used frequently, also in adolescent/student population, with slightly different cut-offs and divided into potential alcohol-related problems into risk levels (see table below).
• What does the use of different cut-offs, and categories in the literature make for comparison between studies?
• Would an AUDIT score of ≥8 for be comparable between studies?
The mentioned studies in the table below are from (Heather et al. (50), Saether et al. (51), Jensen et al. (52), and Tembo et al. (2917), there are likely to be more references.
* Tembo C, Burns S, Kalembo F. The association between levels of alcohol consumption and mental health problems and academic performance among young university students. PLoS ONE. (2017).
Line 239-241: This statement is incorrectly cited/referenced: In the Sub-Saharan study by
Mekonen et al. (53), several components related to substance use (current smoking, chewing khat at least weekly, drinking alcohol on a daily basis, and having intimate friend who uses substance) were all negatively associated with students' academic performance. In
the Finish study by El Ansari et al. (54), academic performance variables (study discipline, importance of achieving good grades, and academic performance compared to peers) were not associated with alcohol consumption behaviors. Consider rewriting.
Line 245: Transitional word “Similarly”, I suggest skip it. “those” should be “male”
Line 253-255: Citation style differs from the rest, keep one style for consistency, consider: “López-Guimerà et al. (61), ...”. Also the mean EEQ score has two decimals while the rest of data presented in the discussion has one decimal. Further, authors are comparing median EEQ with mean values of EEQ the two sentences on line 253-255.
Line 257: Do the authors think that the outcome would have been the different if the study was to be conducted under non-pandemic circumstances? Limitations? Discussion?
Line 262-263: Consider: “In this work, there was a weak positive relationship/correlation between emotional eating and BMI (r=0.15; p=<0.05) in female students.
Line 263: Citation style differs from the rest, keep one style for consistency, consider: “Isik and Cengiz (65)”
Line 265: “like Appraisal Due to Emotions and Stress Questionnaire (EADES) or Three-Factor Eating Questionnaire-R18, which also related strong association..” should be “such as the
Eating and Appraisal Due to Emotions and Stress Questionnaire and the Three-Factor Eating
Questionnaire-R18, relating strong association..”
Line 266: The authors chose to abbreviate the questionnaire “(EADES)” but not the “Three-
Factor Eating Questionnaire-R18” (TFEQ-R18). Please choose on or the other. Or consider rewriting:“Lazarevich et al. (14) and Konttinen et al. (66) found strong associations between emotional eating behaviors and weight gain. However, the studies used different methods to measure emotional eating behavior, such as the Eating and Appraisal Due to Emotions and Stress Questionnaire and the Three-Factor Eating Questionnaire-R18 (14, 66)”
Line 267-273: This is speculative and should be removed or rewritten as food addiction and eating disorders was not in the scope of the authors study.
Line 276: “date” should be “data” “consume” should be “consumption”
Line 277-279: “Another limitation.. ..of the sample” should be ” “Fourthly, there was an imbalance between sexes which could reduce the representativeness of the sample”
On the other hand: authors should investigate the sex distribution of Spanish university
students in health sciences, perhaps the sample is representative?
Line 279: “Forty,“ should be “Fifthly,“ or “Finally,” depending on Line 281 (below).
Line 281: Is the last sentence a limitation too? If so, insert “Finally, “ before “To evaluate...”
Do the authors not think that the manuscript has any strengths?
Line 283-289: The conclusion is blurry..
“In conclusion, this study showed that the university students had a low adherence to the Mediterranean diet, an important high-risk alcohol consumption and low emotional eating. Future co-design work to design e invention strategies focusing on eating behaviors and linked alcohol use for young adults must be developed. On the other hand, must be necessary the implementation of actions to correct deviations of the alcohol abuse and negative food relationship, to avoid unhealthy consequences.”
Do the authors mean?
“In conclusion, this study showed a large proportion of students with low adherence to the Mediterranean diet, a high proportion of students with high-risk alcohol consumption, and eating in response to emotions. Future co-design work to design invention strategies focusing on eating behaviors and alcohol use in young adults must be developed. Furthermore, to
avoid unhealthy consequences implementation of actions to correct deviations of alcohol abuse and negative food relationships is necessary.”

Author Response
Dear reviewer, first, we want to thank you for all the contributions to our manuscript. We are sure that all your suggestions contribute to improve the quality of our study as well as make the dissemination of the results more understandable for the audience interested. Below we detail point by point the changes made to the manuscript after your reviews.
Reviewer 1
- The manuscript would benefit from consistency in the order of the three areas of health research alcohol consumption, adherence to Mediterranean diet and emotional eating.
Deciding the order and following the same pattern consistently throughout would make it easier for the reader to follow
According to the Reviewer considerations, we have modified the order of these three areas: Adherence to Mediterranean diet, alcohol consumption and emotional eating, to make easy the understanding.
- “Alcohol consumption, adherence to Mediterranean diet and emotional eater study in
university students”, could be a little bit more descriptive and precise, consider suggestion: “Alcohol consumption, adherence to the Mediterranean diet and emotional eating in Spanish university students”
We agree with Reviewer and change the title to be more precise.
Abstract:
- Line 10 and 18: “alcohol dependence (SDA)” should be “alcohol dependence syndrome (ADS)”. Line 18: Use only abbreviation ADS, without parentheses, as it has been explained once before in the abstract.
We agree with the Reviewer and we have included the text according the recommendations.
- Line 13: Include “and AUDIT-consumption (AUDIT-C)” after “(AUDIT)” as both questionnaires were used to detect early risk alcohol consumption.
According to the Reviewer suggestion, we added AUDIT-C (line 13).
- Line 14: Insert “(EEQ)” after “Emotional Eater Questionnaire”.
We appreciate the Reviewer suggestion, and we have inserted EEQ.
- have a “Firstly, “and “Secondly, “ etc. describing the findings before. I suggest skip it. Consider following for Line 18-19: “About 40% of the students were categorized as eating very emotionally or eating emotionally, and 35% were categorized as low emotional eaters.”
We appreciate the Reviewer suggestion, and we changed that phrase as Reviewer recommended to the authors.
- Line 22-24: The manuscript would further benefit from a clear conclusion. What do the authors mean with this conclusion? “… to explore the beliefs and behaviors…” ?
We agree with the Reviewer and we have removed the unclear text according the recommendations.
Introduction:
- Line 35-37: Although the authors have cited previously published reporting about unhealthy habits (line 30) and unhealthy eating pattern (line 35) in university student, a brief description regarding the unhealthy habits/eating pattern of university students should be included in the current report (i.e. decreased/or low physical activity, high junk food consumption, evening snacking, high levels of perceived stress, increased workload, dietary restraint and living in residence, and alcohol consumption).
We appreciate the Reviewer suggestion, and we included that brief description in the final manuscript (Line 30-37).
- The introduction would further benefit from a description of the student years/adolescent period of life (i.e. Transition to university is a critical period for young adults. Facing their first opportunity to make their own food choices, having lack of time and money may determine the adoption of unhealthy eating habits and sedentary patterns, leading to increased weight which may persist during adulthood).
We appreciate the Reviewer suggestion, and we included that in the final manuscript.
- Also, a brief overview of what the Mediterranean diet includes or excludes, to help get the reader on track. (i.e. The Mediterranean diet represents a dietary pattern that incorporates healthy traditional eating habits of populations from countries surrounding the Mediterranean Sea. The Mediterranean diet promotes primarily unprocessed foods, including a high consumption of vegetables, fruit, whole grains, legumes, nuts, beans, fish, olive oil, and low consumption of red meat and dairy products).
We appreciate the Reviewer suggestion, and we included a brief overview in the report (Line 62-66).
- Line 58: “energetic foods” suggestion of rephrasing to “high-energy-dense foods”
We appreciate the Reviewer suggestion, and we changed that phrase
Materials and Methods:
- Line 84: “(WHO)” is not needed, as it is not appearing again.; Line 94: Please insert citation/ref for cut-offs; Line 96: Please insert citation/ref for AUDIT
According to the Reviewer suggestion, we have deleted WHO of the manuscript (line 135), we have inserted references for cut-offs reference and AUDIT
- Line 101-104: According to the English translated abstract of reference 23, Garcia Carretero et al. (23), cut-off points for high-risk drinking in students are 8 for male and 6 for female (and 13 for probable ADS, in both sexes). Did the authors use the different cut-offs for the AUDIT between sexes? If so, please clarify in methods. Also, according to the abstract (23) AUDIT-C cut-off points for high-risk patterns for male and female are 5 and 4, respectively. Where these used?
We agree with the comment of the Reviewer and we have added Garcia Carretero et al., cut-off for male and female (Line 148-150).
- Line 102 and 103: Citation style differs from the rest, keep one style for consistency
According to the Reviewer suggestion, we have modified citation style.
- Line 103-104: Abstainers are not mentioned in Materials and Methods but are mentioned in Results line 172. This category should also be explained in Materials and Methods and that they are withdrawn from the calculation of the proportion of students with risky alcohol consumption.
We apologize for the mistake, and we added abstainers in the version of the manuscript (line 153-154).
- Line 107: Revise sentence, suggestion: “Adequate cut-off for AUDIT-C, ≥3 for detecting hazardous drinking, and ≥4 for diagnosable disorders (24).”
We apologize for the mistake, we change the cut off points as the reference of Carretero et al. (line 152).
- Line 109: EEQ is used here for the first time and should be explained before. Emotional Eating Questionnaire (EEQ)
According to the Reviewer suggestion, we added Emotional Eater Questionnaire.
- Line 113: “We classified participants” should be “Participants were classified as:”
Line 115: Citation/ref to classification.
According to the Reviewer suggestion, we have modified the text in line 160 and modified citation in line 162.
- Line 116-117: The statistics presented on the validity of EEQ in Spanish university students, from the cited report Bernabeu et al. (26), are results from the validation of the EEQ-7 item version. According to line 111 the 10-item questionnaire was used. Please revise.
According to the Reviewer suggestion, we have deleted Bernabeu et al in the report and we have added the original validation of the EEQ from Garaulet et al.
- Line 130: Citation/ref to thresholds.
We apologize for the mistake, we added the citation according to the Reviewer suggestion
Line 141-145: The three first sentences can be erased as they repeat data seen in the table. Please correct age data to one decimal and with correct SD as written in the first sentence.
Suggestion of what to leave/rewrite: “Table 1 shows patient characteristics and anthropometry, 26.7% were overweight or had obesity (29.4% male and 27.1% female, p>0.05).”
According to the Reviewer´s suggestions, we have deleted the repeat data and we have included the information suggested in the Table 1 (Line 270).
Table 1
• According to text in (Line 112-113) in section Materials and Methods the authors classified participants accordingly (from a global score range 0 to 30): non-emotional eater (score 0-5), low emotional eater (score 6-10), emotional eater (score 11-20), and very emotional eater (score 21-30). Please use the same classifications in Table 1 for consistency.
As Reviewer suggested, we modified the classification in the Table 1 according to text in section Materials and Methods.
- It would also be helpful to have the scores in the table or table text.
Following the Reviewer´s recommendation, we have included the scores in the table text.
- According to the table text bold values indicate statistical significance. The are no bold values.
We apologize for the mistake. We have deleted this sentence in the table text.
- P-values are shown for age, height, weight, BMI, BMI for underweight, low adherence to MD, and no emotional eating. The rest of the data have no p-values. Does this mean that the data not followed by p-value have the same p-value as the data above? Please clarify.
We appreciate the suggestion of the Reviewer. The p-value refers to the Chi-Square statistical analysis between two categorical variables (BMI, level of adherence to MD and level of emotional eating with sex).
- According to the discussion, line 215-216 “observed less adherence” if this is not significant it should be stated in the discussion.
We agree with the Reviewer, and we have indicated in the discussion that differences in adherence to the Mediterranean diet were not significant (456-458)
The data in Table 1 is not consistent with the results presented in abstract Line 18-19, and with results in text Line 183-185. In the abstract “39.8% of students were very emotional or emotional eater and 35.4% were low emotional eater”, and in the discussion (line 255-256) “38.6% of students were classified as emotional eaters or very emotional eaters”. Please revise all.
We apologize for the mistake. We have changed the mistake of the data in Table 4 and abstract.
Line 154-155: Change sentence to person first language: “Adherence to the Mediterranean diet was similar in participant with normal weight and participants with overweight or obesity (Chi square test, p>0.05)”.
Line 158: Change sentence to person first language: “Women with normal weight were more compliant with the fish consumption recommendation than women with overweight or obesity (43% vs. 33%, respectively, Chi-square test, V=0.09, p<0.05).”
As Reviewer suggested, we have changed the sentence to person first language.
Table 2
- Insert citation/reference to the recommendations, cut-offs for the different recommendation according to MEDAS-14.
According to the Reviewer´s suggestion, we have included the reference to the different recommendations according to MEDAS-14.
Only question 7 and 8 are bold, is that correct?
This is correct. We only observed significant differences between genders for these two items.
Line 167-170: Repeated information as presented in table 3 and can be erased. Instead after the first sentence: “Table 3 shows results from the AUDIT and AUDIT-C questionnaires. Twenty-seven percent reported drinking alcohol 2-3 times/week, 42% reported drinking alcohol 2-4 times/month, and 8.6% reported being abstainers (with no difference between sex, p>0.05, for all).”
Line 174-176: Repeated information as presented in table 3 and can be erased, or only keep text (without data) and refer to Table 3. If kept: Line 174-175: “higher” should be “lower” and “high-risk consumption” should be “low-risk consumption”, according to results in Table 3.
Following the Reviewer´s recommendation, we have removed the repeated information and we have rewritten that paragraph.
According to Table 3, a higher proportion of women (37.4%) had high-risk consumption compared to men (26.0%).
Line 176: Transition word “Similarly” I suggest, skip it. Consider rewriting sentence accordingly:“Men with high-risk alcohol consumption had lower adherence to the Mediterranean diet (Chi square test, X2=7.05, V=0.11, p<0.05) (Table 3).”
As Reviewer suggested, we have removed the transition word and we have modified the sentence according to your recommendations.
Line 178: Change sentence to person first language: “There was no difference in alcohol consumption between students with normal weight and those with overweight or obesity (p>0.05).”
Following the Reviewer´s recommendation, we have changed the sentence to person first language.
Table 3.
- Explain/insert cut-offs for AUDIT and AUDIT-C, preferably in the table or table text.
According to Materials and Methods (line 107) and cited Reinert et al. (24), AUDIT-C cut-off ≥3 is for detecting hazardous drinking, and ≥4 is for diagnosable disorders. Are these interpreted as “≥3 = low-risk consumption” and “≥4 = high-risk consumption”?
If so, this should be explained in Materials and Methods. If not, revise text and table to correct terms/interpretations
We appreciate the comment. According to Reinert et al. (24), AUDIT-C cut-off for identifying hazardous drinking is different according to gender (≥3 in women and ≥4 in men). We have specified the criteria used in Material and Methods and we have included cut-offs for AUDIT and AUDIT-C in table text.
P-values are shown for AUDIT Low-risk drinkers and AUDIT-C Low-risk consumption.
The rest of the data have no p-values. Does this mean that the data not followed by p-value have the same p-value as the data above? Please clarify.
We appreciate the suggestion of the Reviewer. The p-value refers to the Chi-Square statistical analysis between two categorical variables (sex and AUDIT/AUDIT-C).
Line 182-185: Keep first sentence. Second sentence is a repetition of what is presented in table 1. Suggestion after first sentence: “Table 1 shows proportions of students scoring low emotional eaters, emotional eaters, and very emotional eaters. 3X.X% of students were classified as emotional or very emotional eaters according to EEQ (Table 1.).”
Following the Reviewer´s recommendation, we have removed the repeated information and we have rewritten this paragraph.
Line 186: Skip the first transitional word “Finally” and start the sentence with “An association...”.
As Reviewer suggested, we have removed the first transition word.
Line 191-192: A correlation value of 0.15 shows there is a positive correlation between two variables, but it is weak and likely unimportant, this should be outlined. Line 19-20, 191-192, 262-263
We appreciate the suggestion of the Reviewer. We indicated the thresholds used to interpret the correlation coefficients in the line 189-191. Following the reviewer's recommendation and to facilitate the understanding of the results, we have included the degree of correlation in each of the coefficients that appear in the results.
Discussion
Line 200: “date” should be “data”
Line 201: “others” should be “other Spanish university student populations”.
Line 203: Transitional phrase “In the same way”should be “Furthermore”
Line 204: Authors have missed to write “students” after “university” before “with”.
Line 210: “an improved” should be “improvement” plus insert “in adults” at the end of the sentence.
Line 211: Sentence correction suggestion: “The present study shows that a large proportion of university students had low adherence to the Mediterranean diet.”
Line 212-214: Citation style differs from the rest, keep one style for consistency, consider: “Cobo-Cuenca et al. reported the... ... the Mediterranean diet (30), similar with....”
Line 219: “haven’t” should be “have not”.
Line 223: “(39% men and 39% women” should be “(39% both in sexes)”
Line 225-227: Sentence correction, and citation style differs from the rest, keep one style for consistency: “As a matter of fact, Cobo-Cuenca et al. reported that only 30% male and 32% female Spanish university students had fish and shellfish in agreement with the
Mediterranean diet recommendation (30).”
Line 228: Change sentence to person first language: “female university students with overweight or obesity...”
Line 233: “as” should be “than”.
Line 234: Citation style differs from the rest, keep one style for consistency, consider: “...
by Cooke et al. (48), and...” Or consider following for Line 233-234: “The median AUDIT score of 5.3±4.4 observed in the present study, is similar to a previous report in Spanish university students (49), but lower than observed in other European countries
(with median AUDIT score of 6.9 ± 5.5) (48).”
Line 235: Please insert citation/reference 23 in“According to risk categories (23), more than...”
Line 245: Transitional word “Similarly”, I suggest skip it. “those” should be “male”
Line 263: Citation style differs from the rest, keep one style for consistency, consider: “Isik and Cengiz (65)”
Line 265: “like Appraisal Due to Emotions and Stress Questionnaire (EADES) or Three-Factor Eating Questionnaire-R18, which also related strong association..” should be “such as the Eating and Appraisal Due to Emotions and Stress Questionnaire and the Three-Factor Eating Questionnaire-R18, relating strong association..”
Line 266: The authors chose to abbreviate the questionnaire “(EADES)” but not the “Three-Factor Eating Questionnaire-R18” (TFEQ-R18). Please choose on or the other. Or consider rewriting:“Lazarevich et al. (14) and Konttinen et al. (66) found strong associations between emotional eating behaviors and weight gain. However, the studies used different methods to measure emotional eating behavior, such as the Eating and Appraisal Due to Emotions and Stress Questionnaire and the Three-Factor Eating Questionnaire-R18 (14,66)”.
Line 276: “date” should be “data” “consume” should be “consumption”
Line 279: “Forty,“ should be “Fifthly,“ or “Finally,” depending on Line 281 (below).
Line 281: Is the last sentence a limitation too? If so, insert “Finally, “ before “To evaluate...”
We apologize for the mistake. We have checked carefully the document and we have improved the writing in the final version of the manuscript.
Line 236-237: This statement is incorrectly cited/referenced. In the study by Heather et al.,(2011) 40% of adolescents were hazardous drinkers (AUDIT score 8-15), 11% harmful drinkers (AUDIT score 16-19), and 10% probable dependence (ADS) (AUDIT 20 or over). In the present study 26.2% scored high-risk drinkers (AUDIT score 8-12) and 7.7% scored probable dependence (ADS) (AUDIT score 13 and over). Over 30% (33.9%) of the university students in this present study had high-risk consumption and/or ADS (Audit score 8 and above). If correctly compared with the study by Heather et al. AUDIT scores of 8 and above would be 61% of the adolescents. If the authors meant to compare the prevalence of adolescents with probable dependence (ADS), 7.7% in the present study with 10% by Heather et al. The cut-off points for probable dependence (ADS) assessed by AUDIT score of 13 and above vs. 20 and above would have to be addressed and discussed.
We agree with the Reviewer and we have highlighted the different cut-off points for assessing ADS compared to Heather et al. (Line 478-480)
On that note, AUDIT is used frequently, also in adolescent/student population, with slightly different cut-offs and divided into potential alcohol-related problems into risk levels (see table below).
- What does the use of different cut-offs, and categories in the literature make for comparison between studies?
- Would an AUDIT score of ≥8 for be comparable between studies?
We agree with the Reviewer that there are different cut-off points for interpreting AUDIT results. However, the table shows that most studies identify a score above 8 as risky consumption and lower values as low risk. Therefore, the results may be of interest for comparing risky consumption between studies.
Line 239-241: This statement is incorrectly cited/referenced: In the Sub-Saharan study by
Mekonen et al. (53), several components related to substance use (current smoking, chewing khat at least weekly, drinking alcohol on a daily basis, and having intimate friend who uses substance) were all negatively associated with students' academic performance. In the Finish study by El Ansari et al. (54), academic performance variables (study discipline, importance of achieving good grades, and academic performance compared to peers) were not associated with alcohol consumption behaviors. Consider rewriting.
Following the recommendations of the Reviewers we have modified the references used according to the statement (Line 485).
Line 253-255: Citation style differs from the rest, keep one style for consistency, consider: “López-Guimerà et al. (61), ...”. Also the mean EEQ score has two decimals while the rest of data presented in the discussion has one decimal. Further, authors are comparing median EEQ with mean values of EEQ the two sentences on line 253-255.
As Reviewer suggested, we have modified the citation style.
Regarding the comment related to the EEQ score, we commented that the mean EEQ score in our study was lower than previous studies (Line 561). We consider that these differences could be caused by the age range of the study participants.
Line 257: Do the authors think that the outcome would have been the different if the study was to be conducted under non-pandemic circumstances? Limitations? Discussion?
We understand the appreciation of the Reviewer, This study was carried during 2018-2019 under non-pandemic circumstances (Line 98)
Line 262-263: Consider: “In this work, there was a weak positive relationship/correlation between emotional eating and BMI (r=0.15; p=<0.05) in female students.
According to the Reviewer´s suggestion, we have modified this statement (Line 571)
Line 267-273: This is speculative and should be removed or rewritten as food addiction and eating disorders was not in the scope of the authors study.
We agree with the Reviewer and we decided remove this sentence
Line 277-279: “Another limitation.. ..of the sample” should be ” “Fourthly, there was an imbalance between sexes which could reduce the representativeness of the sample”
We agree with the Reviewer and we decided remove this sentence.
On the other hand: authors should investigate the sex distribution of Spanish university
students in health sciences, perhaps the sample is representative?
In Spain, the majority of university students in health sciences are women. Following the Reviewer´s recommendation, we have explained this issue (Line 585-587)
Do the authors not think that the manuscript has any strengths?
As Reviewer suggested, we have included the strength of the manuscript. (Line 590-593)
Line 283-289: The conclusion is blurry. “In conclusion, this study showed that the university students had a low adherence to the Mediterranean diet, an important high-risk alcohol consumption and low emotional eating. Future co-design work to design e invention strategies focusing on eating behaviors and linked alcohol use for young adults must be developed. On the other hand, must be necessary the implementation of actions to correct deviations of the alcohol abuse and negative food relationship, to avoid unhealthy consequences.”
Do the authors mean?
“In conclusion, this study showed a large proportion of students with low adherence to the Mediterranean diet, a high proportion of students with high-risk alcohol consumption, and eating in response to emotions. Future co-design work to design invention strategies focusing on eating behaviors and alcohol use in young adults must be developed. Futhermore, to avoid unhealthy consequences implementation of actions to correct deviations of alcohol abuse and negative food relationships is necessary.”
Following the Reviewer´s recommendation, we have modified the conclusions of the present work.

Reviewer 2 Report
The manuscript submitted for publication titled: Alcohol Consumption, Adherence to Mediterranean Diet and Emotional Eater Study in University Students.
The reviewer would like to present the following points regarding the manuscript.
Abstract
- Abstract needs to be improved. The number of subjects described in the abstract is different from the number of subjects presented in the method or results. Also, it is different from the result description of the result part or the contents of the table. Numbers are very important, so be consistent with accurate results.
Introduction
- The introduction needs to provide more specific evidence as per the novelty of the work conducted and its contribution to the body of knowledge in the field and demonstrate how it helps the field advance. Relevant statements should be included.
- The authors analyzed the statistical analysis according to sex. If sex is an important consideration, please add need or background to it.
- The research question should be included in at the end of the introduction section.
Material and Methods
- Please specify the data collection period by month.
- Please check the bibliography citation method, such as Carretero et al (2016).
Results
- Since this is a cross-sectional study and the related contents are described in the main text, Figure 1 can be deleted.
- There are some differences in the number of participants and results in the description of the abstract, table, and text. Please check and make consistent corrections.
- Only the p-values greater than or less than 0.05 are presented in the table. Please provide a specific p-value. No statistics are presented in the table. Please add statistics.
- All results should be described in tables, and meaningful ones should be described in the text. However, there are parts in the text that are not in the table. If the author wish to describe it in the text, please also add relevant information to the table. In particular, is it meaningful to present each item in Table 2? The mean and standard deviation are presented in the main text, but the table does not contain relevant information.
Discussion
- What is described in the discussion must be presented in the results. In this study, the mean and standard deviation of Audit or the mean and standard deviation of EEQ are discussed as specific numbers in the discussion. However, there is no relevant content in the result section, so it is better to add related results.
- The discussion is somewhat limited and does not seem to consider broadly the literature in response to the findings of the study. Furthermore, it does not seem to adequately position the findings and provide concrete recommendations and future directions for research.
Author Response
Dear reviewer, first, we want to thank you for all the contributions to our manuscript. We are sure that all your suggestions contribute to improve the quality of our study as well as make the dissemination of the results more understandable for the audience interested. Below we detail point by point the changes made to the manuscript after your reviews.
Reviewer 2
Abstract:
The number of subjects described in the abstract is different from the number of subjects presented in the method or results. Also, it is different from the result description of the result part or the contents of the table. Numbers are very important, so be consistent with accurate results
We apologize for the mistake. We have changed the number of subject described in the abstract.
Introduction:
The introduction needs to provide more specific evidence as per the novelty of the work conducted and its contribution to the body of knowledge in the field and demonstrate how it helps the field advance. Relevant statements should be included.
Following the Reviewer´s recommendation, we have modified the introduction of the final version of the manuscript.
The authors analyzed the statistical analysis according to sex. If sex is an important consideration, please add need or background to it.
We agree with the Reviewer, and we have added this information in the introduction (line 88-90)
The research question should be included in at the end of the introduction section
We have included the question as the reviewer pointed (line 91-92)
Material and Methods
Please specify the data collection period by month.
Regarding the comment related to data collection we have indicated the date collection by month (line 97-98)
Please check the bibliography citation method, such as Carretero et al (2016).
We apologize for the mistake, we have modified the citation method
Results
Since this is a cross-sectional study and the related contents are described in the main text, Figure 1 can be deleted.
According to the Reviewer´s suggestions, we have deleted the Figure 1 of the final version of the manuscript.
There are some differences in the number of participants and results in the description of the abstract, table, and text. Please check and make consistent corrections.
We apologize for the mistake. We have checked carefully the document and we have improved the writing in the final version of the manuscript.
Only the p-values greater than or less than 0.05 are presented in the table. Please provide a specific p-value. No statistics are presented in the table. Please add statistics.
We appreciate the Reviewer suggestion and we have provided a specific p-value in table 1 and 3
With respect to the statistics, we have added the statistics used in table 1.
All results should be described in tables, and meaningful ones should be described in the text. However, there are parts in the text that are not in the table. If the author wish to describe it in the text, please also add relevant information to the table. In particular, is it meaningful to present each item in Table 2? The mean and standard deviation are presented in the main text, but the table does not contain relevant information.
Following the recommendations of Reviewer 1, we have removed the parts of the text as repeat data seen in the tables.
Regarding the comment related to table 2, we believe it is important to present each element of the table in order to identify which dietary factor it refers and to be able to interpret the results obtained.
In table 2 the values are presented in percentages according to the degree of compliance for each item while the main text describes the average MEDAS score and the degree of adherence of the population.
What is described in the discussion must be presented in the results. In this study, the mean and standard deviation of Audit or the mean and standard deviation of EEQ are discussed as specific numbers in the discussion. However, there is no relevant content in the result section, so it is better to add related results.
We appreciated the Reviewer comment. The mean and standard deviation of Audit is mentioned in results (line 350). Likewise, the mean and standard deviation of EEQ is mentioned in results (lines 399).
The discussion is somewhat limited and does not seem to consider broadly the literature in response to the findings of the study. Furthermore, it does not seem to adequately position the findings and provide concrete recommendations and future directions for research
We appreciate the Reviewer suggestion. In the discussion we have considered the most relevant scientific literature in relation to the findings obtained in this paper, but following the reviewer's suggestions we have completed the discussion.
Following the Reviewer´s recommendation and base on the results obtained in this study, we have included in the conclusion paragraph, some recommendations and future directions for research.
